# CRISPR-Cas9-Mediated Cytosine Base Editing Screen for the Functional Assessment of *CALR* Intron Variants in Japanese Encephalitis Virus Replication

**DOI:** 10.3390/ijms241713331

**Published:** 2023-08-28

**Authors:** Youcai Xiong, Xiaoning Xi, Yue Xiang, Sheng Li, Hailong Liu, Yinyu Su, Ruigao He, Chong Xiong, Bingrong Xu, Xinyi Wang, Liangliang Fu, Changzhi Zhao, Xiaosong Han, Xinyun Li, Shengsong Xie, Jinxue Ruan

**Affiliations:** 1Key Laboratory of Agricultural Animal Genetics, Breeding and Reproduction of Ministry of Education & Key Laboratory of Swine Genetics and Breeding of Ministry of Agriculture and Rural Affairs, Huazhong Agricultural University, Wuhan 430070, China; youcaixiong@webmail.hzau.edu.cn (Y.X.); xiaoningxi@webmail.hzau.edu.cn (X.X.); yxiang@webmail.hzau.edu.cn (Y.X.); lisheng@webmail.hzau.edu.cn (S.L.); hailongliu@webmail.hzau.edu.cn (H.L.); 2020302110037@webmail.hzau.edu.cn (Y.S.); heruigao@webmail.hzau.edu.cn (R.H.); xc0909@webmail.hzau.edu.cn (C.X.); brxu@webmail.hzau.edu.cn (B.X.); wangxinyi2019@webmail.hzau.edu.cn (X.W.); fuliangliang2017@mail.hzau.edu.cn (L.F.); czzhao@mail.hzau.edu.cn (C.Z.); xshan@mail.hzau.edu.cn (X.H.); xyli@mail.hzau.edu.cn (X.L.); 2The Cooperative Innovation Center for Sustainable Pig Production, Huazhong Agricultural University, Wuhan 430070, China; 3Hubei Hongshan Laboratory, Frontiers Science Center for Animal Breeding and Sustainable Production, Wuhan 430070, China

**Keywords:** base editing, *CALR*, Japanese encephalitis virus, saturation point mutation screening

## Abstract

The Japanese encephalitis virus (JEV) is a mosquito-borne flavivirus that causes viral encephalitis in humans, pigs and other mammals across Asia and the Western Pacific. Genetic screening tools such as CRISPR screening, DNA sequencing and RNA interference have greatly improved our understanding of JEV replication and its potential antiviral approaches. However, information on exon and intron mutations associated with JEV replication is still scanty. CRISPR-Cas9-mediated cytosine base editing can efficiently generate C: G-to-T: A conversion in the genome of living cells. One intriguing application of base editing is to screen pivotal variants for gene function that is yet to be achieved in pigs. Here, we illustrate that CRISPR-Cas9-mediated cytosine base editor, known as AncBE4max, can be used for the functional analysis of calreticulin (*CALR*) variants. We conducted a CRISPR-Cas9-mediated cytosine base editing screen using 457 single guide RNAs (sgRNAs) against all exons and introns of *CALR* to identify loss-of-function variants involved in JEV replication. We unexpectedly uncovered that two enriched sgRNAs targeted the same site in intron-2 of the *CALR* gene. We found that mutating four consecutive G bases in the intron-2 of the *CALR* gene to four A bases significantly inhibited JEV replication. Thus, we established a CRISPR-Cas9-mediated cytosine-base-editing point mutation screening technique in pigs. Our results suggest that CRISPR-mediated base editing is a powerful tool for identifying the antiviral functions of variants in the coding and noncoding regions of the *CALR* gene.

## 1. Introduction

Japanese encephalitis virus (JEV) is a mosquito-transmitted virus that can infect humans and livestock, and thus JEV poses a threat to human public health and livestock growth and development [1,2]. JEV has spread across vast geographical areas such as India, China, and almost all of Southeast Asia, with approximately 50,000 cases of JEV infection reported worldwide each year and about 20% of cases resulting in death [3]. In addition, JEV can rapidly proliferate in pigs, thus leading to high levels of viremia [4].

Genetic screening tools such as CRISPR screening, DNA sequencing, RNA interference (RNAi) screening and microarrays have greatly improved our understanding of JEV replication and its potential antiviral approaches [5]. In vitro-directed evolution of candidate protein is a highly effective and vastly functional framework for altering the activities of individual genes and gene products using tools such as error-prone PCR and DNA editors to generate mutations and direct protein evolution [6,7,8]. However, these tools are inefficient, hence the need for carefully developed methods to screen and select functional variants in manners that maintain the genotype–phenotype association [9]. Cytosine base editing (CBE) typically enables the insertion of specific point mutations generating C:G-to-T:A conversion on the target DNA without inducing double-strand breaks (DSBs) [10]. CRISPR-mediated cytosine base editing (CBE) methods are commonly used in the biological and agricultural fields primarily for saturation screening of amino acids [10,11,12]. A promising application of base editing is the in vivo screening for mutations crucial in protein function where critical amino acids closely related to the phenotype are identifiable through base-editing saturation screening [13,14]. It is possible to adopt current base-editor methods to screen for the functional mutations in the non-coding gene regions, such as introns, enhancers and promoters that may have significant roles in the physiological processes [15].

The calreticulin (*CALR*) gene is an essential host factor for JEV replication in pigs, but the gene’s homozygous knockout will cause embryo death, limiting its in depth application [5,16,17]. To overcome this obstacle, we hypothesized that introducing mutations in this gene might allow cells to inhibit JEV replication without affecting the expression of the CALR protein. Mutations within the exons or introns of the *CALR* gene can inhibit JEV, which is of great significance in further revealing the mechanisms of JEV replication. Here, we identified the *CALR* mutations critical for JEV replication in porcine kidney-15 (PK-15) cells by employing CRISPR-based cytosine base editing screening and an optimized sgRNA library covering all exons and introns of the *CALR* gene. Our optimization strategy of the sgRNA library provides an alternative method for the in vitro screening of more functional mutations. Our CRISPR-based cytosine-base-editing screening resources are valuable for mutation generation and directed evolution in structural and functional genomics studies.

## 2. Materials and Methods

### 2.1. Plasmids

The plasmids PB-EF1ɑ-ancBE4max-pGK-puro, piggyBac transposase and Lenti-sgRNA-EGFP were provided in kind by Professor Xingxu Huang at Shanghai Technology University. The PB-EF1ɑ-ancBE4max-pGK-puro and piggyBac transposase plasmids were used to generate the AncBE4max-expression cell line (PK15-BE4). The lentiviral sgRNA vector was constructed by annealing the paired sgRNA oligonucleotides and cloning them into Lenti-sgRNA-EGFP linearized with *Bbs*I (NEB, Ipswich, MA, USA).

### 2.2. Design and Construction of sgRNA Library

The sgRNAs covering all the exons and introns of the *CALR* gene were designed using the CRISPR-offinder software (https://sourceforge.net/projects/crispr-offinder-v1-2/ (accessed on 8 October 2019)) [18]. The designed sgRNA sequences are listed in Appendix A. Subsequently, the sgRNAs were synthesized (CustomArray Inc., Redmond, WA, USA) and amplified (20 PCR cycles) using Phusion high-fidelity PCR Master Mix (NEB, Ipswich, MA, USA) according to the manufacturer’s instructions. The PCR products were purified using the MinElute PCR purification Kit (QIAGEN, Dusseldorf, Germany) and ligated into the linearized Lenti-sgRNA-EGFP vector using Gibson assembly cloning kit (NEB, Ipswich, MA, USA). Trans1-T1 Phage Resistant chemically competent cells (TransGen Biotech, Beijing, China) were transformed using ligation products. The transformations were conducted in parallel to ensure sufficient coverage and enable the colony numbers to reach 200 times the total sgRNA number in the library. Afterwards, sgRNA library plasmids were extracted using a Plasmid Plus Maxi Kit (QIAGEN, Dusseldorf, Germany) and amplified using Phusion high-fidelity PCR Master Mix (NEB, Ipswich, MA, USA) with 20 cycles. The resultant PCR products were purified and analyzed by high-throughput sequencing. The PCR and sequencing primers used are listed in Appendix A.

### 2.3. Cell Culture and Transfection

PK-15, HEK293T and BHK-21 cell lines purchased from the Cell Bank of the Chinese Academy of Sciences (Shanghai, China) were cultured in Hyclone Dulbecco’s Modified Eagle Medium (DMEM) supplemented with 10% fetal bovine serum (Hyclone, Logan, UT, USA) and 1% penicillin/streptomycin (Life Technology, Rockville, MD, USA) at 37 °C with 5% CO_2_. PK-15 cells were seeded into 6-well plates, cultured to 70–80% confluence and transfected with 2μg plasmid DNA using the JetPRIME transfection reagent (PolyPlus, Strasbourg, France).

### 2.4. Generation of BE4-Expression Cell Line

The PK-15 cells were cotransfected with PB-EF1ɑ-ancBE4max-pGK-puro and piggyBac Transposase vectors (at ratio = 2:1). At 48 h post-transfection, the culture medium was replaced with fresh media containing 2 μg/mL puromycin (InvivoGen, Toulouse, France). After enrichment for 2 days, antibiotic-resistant cells were collected and reseeded into 100 mm dishes at a density of 100 cells/dish to generate single-cell clones. After being cultured for 7–10 days, the cell clones were singly collected and examined by PCR. In addition, two sgRNAs were used to determine the activity of the selected cell line. The resulting cell line was designated as PK15-BE4.

### 2.5. Preparation and Transduction of sgRNA Library Lentivirus 

The HEK293T cells were cultured and cotransfected with 12 μg library plasmid, 4 μg pMD2.G plasmid (Addgene, Cambridge, MA, USA) and 8 μg psPAX2 (Addgene, Cambridge, MA, USA) plasmid per 100 mm dish using JetPRIME for the preparation of lentivirus. At 60 h post-transfection, the supernatants were collected, filtered through a 0.45 μm low protein-binding membrane (Millipore, Darmstadt, Germany), and centrifuged at 153,700× *g* for 2.5 h at 4 °C. The obtained lentivirus pellets were resuspended in phosphate-buffered saline (PBS, pH = 7.4), aliquoted and stored at −80 °C. Target cells were transduced with the lentiviruses in the presence of 8 μg/mL polybrene (Sigma–Aldrich, St. Louis, MO, USA). At 24 h post-transduction, lentiviruses were removed, and the medium was replaced with a fresh one.

### 2.6. Mutant Cell Library Construction and Screening

Approximately 1 × 10^7^ PK15-BE4 cells were seeded into T75 flasks and infected with the library lentiviruses at a multiplicity of infection (MOI) = 0.3. On the third day post-infection (3 dpi), GFP-positive cells were collected by fluorescence-activated sorting (FACS) on a BD-FACSAria^TM^ II platform. At day 6 postinfection, DNA from mutant cells was extracted and amplified to examine the coverage of the mutant cell library. 

For the CRISPR base-editing screening, 3 × 10^6^ mutant cells were infected with JEV-RP9 strain at an MOI of 1 in DMEM containing no FBS at 37 °C with 5% CO_2_. At 1.5 h postinfection (hpi), the inoculum was removed, and the medium was replaced with fresh DMEM supplemented with 2% FBS and 1% penicillin/streptomycin. On day 20 postinfection, the DNA of viable cells was extracted and amplified for subsequent deep sequencing.

### 2.7. Illumina Sequencing of sgRNAs from the Library and Enriched Mutants

The PCR products were purified with a MinElute PCR purification Kit (QIAGEN, Dusseldorf, Germany) and sequenced by Illumina HiSeq 3000 next-generation sequencing. Mapped reads were subjected to the MAGeCK analysis [19].

### 2.8. Mutation of Candidate Region in PK15 Cells by Base Editing

Individual sgRNAs targeting candidate mutation sites were cloned into the linearized Lenti-sgRNA-EGFP vector to produce the corresponding lentiviruses described in Section 2.5, followed by transduction into PK15-BE4 cells. The GFP-positive cells were enriched by FACS and seeded into 100 mm dishes to generate single clones. The genomic DNAs of single clones were extracted and sequenced. All primers used for identifying the genotype of cell clones are in Appendix A.

### 2.9. Off-Target Analysis of sgRNA

Potential sgRNA off-target sites (OTS) were predicted using the Cas-OFFinder software (http://www.rgenome.net/cas-offinder/ (accessed on 15 July 2021)) [20]. After PCR amplification of each potential OTS, Sanger sequencing was performed to determine whether any point mutation occurred.

### 2.10. Real-Time Reverse Transcription PCR (qRT-PCR)

Total RNA was extracted from cells using TransZol Up (TransGen Biotech, Beijing, China). The quality and quantities of the resultant RNAs were evaluated by gel electrophoresis and the NanoDrop ND2000^TM^ spectrophotometer (ThermoFisher Scientific, Waltham, MA, USA), respectively. The cDNAs were synthesized using the PrimeScript™ RT reagent Kit with gDNA Eraser (TaKaRa, Otsu, Japan). Then, qRT-PCR was performed by RealUniversal SYBR Green Premix (TIANGEN, Beijing, China) following the manufacturer’s instructions.

### 2.11. Western Blot

The cells were collected and lysed in radioimmunoprecipitation assay (RIPA) buffer (Sigma-Aldrich, St. Louis, MO, USA) supplemented with 1% Phenylmethanesulfonyl fluoride (PMSF) (Sigma-Aldrich, St. Louis, MO, USA) and 1% sodium vanadate (Sigma-Aldrich, St. Louis, MO, USA), and proteins quantified with a BCA kit (Beyotime, Shanghai, China). After denaturation, the samples were separated by 12% sodium dodecyl-sulphate polyacrylamide gel electrophoresis (SDS-PAGE) and transferred to the polyvinylidene difluoride membranes (Millipore, Darmstadt, Germany). The polyvinylidene difluoride membranes were blocked with 5% nonfat milk and incubated with antibodies diluted to the appropriate concentration. Next, the membranes were washed with tris-buffered saline with Tween^®^ 20 (TBST) (Sigma-Aldrich, St. Louis, MO, USA) and incubated with H-conjugated antirabbit IgG diluted with 5% nonfat milk. Fluorescence images were captured using GelDoc XR (Bio-Rad, Hercules, CA, USA) and the ChemiDoc XR imaging system (Bio-Rad, Hercules, CA, USA).

### 2.12. Virus Plaque Assay

For the plaque assay, BHK-21 cells were cultured to 50% confluence in 6-well flat-bottomed cell culture plates and infected with the serially diluted virus for 2 h at 37 °C with 5% CO_2_. The inoculum was aspirated, and the cells overlaid with 50% low melt-point Agarose (ThermoFisher Scientific, Waltham, MA, USA) in 2 × DMEM supplemented with 2% FBS and 1% penicillin/streptomycin and cultured for 3 days at 37 °C with 5% CO_2_. Cells were fixed with 10% formaldehyde neutral solution overnight at room temperature, then stained with 0.5% crystal violet for 2 h. The plaque-forming units (PFUs) were determined by manual counting and expressed as means ± SD of 3 independent experiments.

### 2.13. Transmission Electron Microscopy

Approximately 1 × 10^6^ of Mut-1 and WT PK15-BE4 cells were infected with JEV-RP9 (MOI = 1) for 24 h. The cells were washed twice with precooled PBS and fixed by adding 3 mL of fixative (Servicebio Technology, Wuhan, China) for 2 h at room temperature. After fixation, the cells were scraped, transferred into a 1.5 mL centrifuge tube, and centrifuged at 251.55× *g* for 5 min at 4 °C. The images were captured using negative-staining electron microscopy by Servicebio company.

### 2.14. CUT&Tag Analysis

CUT&Tag (H3K4me3 and H3K27ac) data were derived from our unpublished study. Adaptors and low-quality reads were removed by Trim Galore (v0.6.6). The processed reads were aligned to the Sscrofa11.1 pig reference genome using bowtie2 (v2.3.4.3) [21]. MACS2 (v2.2.4) was used to call narrow peaks for H3K4me3 and H3K27ac [22]. Bigwigs were generated using MACS2, BEDTools (v2.29.0) and bedGraphToBigWig and then visualized using Integrative Genomics Viewer (IGV) (v2.9.2) [23,24].

### 2.15. Statistical Analysis

Statistical analysis was performed using R package. The data were expressed as means ± SD. Two-tailed Student’s *t*-test was performed to determine significant differences between treatment and control groups (* *p* < 0.05; ** *p* < 0.01; *** *p* < 0.001; **** *p* < 0.0001).

## 3. Results

### 3.1. Construction of a CRISPR-Cas9-Mediated Base Editing Library for Screening Functional Variants of CALR Gene

*CALR* is an endoplasmic reticulum (ER) resident protein critical for JEV replication. We identified *CALR* functional mutations involved in JEV replication by developing a CRISPR-Cas9-mediated cytosine base editing strategy to perform unbiased single-base saturation screening against all exons and introns of the *CALR* gene (Figure 1a). We first generated a PK-15 cell line expressing AncBE4max (PK15-BE4) with a PiggyBac transposon-mediated gene transfer approach (Figure 1b). After selecting clones, we amplified the sequence of Cas9 and identified the positive clones (Appendix A). The No.7 of PK15-BE4 cell line was selected to further evaluate the editing activity with two sgRNAs targeting the *SLC35B2* and *DRG1* sites (Figure 1c). We chose the PK15-BE4 cell line for further study. Then, 520 sgRNAs containing 457 specific sgRNAs targeting all exons and introns of the *CALR* gene and 63 negative control sgRNAs were designed using CRISPR-offinder (Figure 1d). Next, these sgRNAs were synthesized as an oligo array and cloned into lentiviral vectors using Gibson assembly (Figure 1e).

The cloned sgRNA constructs were PCR amplified and deep sequenced to detect the abundance of the library. The results showed that 98.65% (513/520) of the initially designed and synthesized sgRNAs were in the plasmid library (Appendix A). To further assess the quality of this sgRNA library and determine the optimal screening time for the challenge of JEV, four of the designed sgRNAs were randomly selected and evaluated for their gene-editing activity at different time points. The results showed that the gene-editing activity of these sgRNAs was about 20–60% and stabilized at approximately 10 days postinfection with sgRNA-harboring lentivirus in PK15-BE4 cells (Figure 1f). The results imply that the constructed sgRNA library can be used for subsequent experiments.

### 3.2. Screening for CALR Functional Mutations Involved in JEV Replication by a CRISPR-Cas9-Mediated Cytosine Base Editing Strategy

Subsequently, a single-base saturable mutant cell pool of the *CALR* gene was constructed by infecting PK15-BE4 cells with sgRNA lentivirus (Appendix A). Deep sequencing results showed that most of the initially designed sgRNA sequences were retained in the mutant cell pool (Appendix A). Next, this high-quality mutant cell pool was challenged with JEV at an MOI of 0.1, and the surviving mutant cells were collected and deep sequenced. Through bioinformatics analysis, we obtained a batch of enriched sgRNAs (Figure 1g). And the top three ranked sgRNAs were found to be sg*CALR*_S_215, sg*CALR*_A_521 and sg*CALR*_A_488, respectively (Appendix A).

Then, these top three ranked sgRNAs were further selected for functional validation. Using CRISPR single-base pooled editing experiments, we found that sg*CALR*_A_521 lentivirus infection of PK15-BE4 cells significantly inhibited JEV replication; however, two other enriched sgRNA infected PK15-BE4 cells failed to inhibit JEV replication (Appendix A). Interestingly, we found that sg*CALR*_A_521 targets intron 2 of the *CALR* gene. The results imply that mutations in intron 2 of the *CALR* gene can inhibit viral replication.

### 3.3. Consecutive Mutations of Four Bases in CALR Intron 2 Significantly Inhibit JEV Replication

To further determine the genotype of the intron-2 mutation in the *CALR* gene and to clarify its role, we constructed three cell lines with single-clonal-origin of the *CALR* intron-2 mutation by AncBE4max-mediated base editing technique. Sanger sequencing results showed that only one of these mutant cell lines was homozygous with four consecutive G-base mutations in *CALR* intron-2 (Mut-1). The other two mutant cell lines were heterozygous (Figure 2a and Appendix A). Therefore, the Mut-1 cell line was selected and challenged with JEV. The results showed that Mut-1 cells significantly inhibit the proliferation ability of JEV after 12 h and 24 h of viral infection compared to wild-type cells by qRT-PCR assay (Figure 2b). Using plaque assay, we found that *CALR* Mut-1 cells could inhibit JEV replication, which was consistent with the qRT-PCR assay results (Figure 2c).

The density of viral particles in Mut-1 and wild-type cells was assessed with Negative Staining Electron Microscopy. The results showed that viral particles were observed in the endoplasmic reticulum of JEV-infected wild-type PK-15 cells but not in the endoplasmic reticulum of Mut-1 cells (Figure 2d). In addition, our data showed the diminution of mitochondrial crista in wild-type PK-15 cells but no diminution in Mut-1 cells after JEV infection (Figure 2d). By evaluating the predicted off-target sites of sgRNA, we found that the constructed Mut-1 mutant cells had relative high specificity (Appendix A). Next, we examined CALR protein expression in Mut-1 cells. Unexpectedly, its protein expression was unchanged in the *CALR* mutant cells compared to wild-type cells (Figure 2e). Moreover, we found that Mut-1 cells can suppress the expression of JEV-encoded NS3 (Figure 2e). In addition, the CUT&Tag analysis showed that H3K4me3 and H3K27ac histone signals were detected in the mutation region (Figure 2f), implying that the intron-2 mutation site might be a transcription factor binding site, requiring further exploration. Taken together, we found that the mutation of four consecutive G bases to four A bases in intron 2 of the *CALR* gene can inhibit JEV replication.

## 4. Discussion

We identified an intron mutation crucial for JEV replication using CRISPR-base-editing screening based on an optimized sgRNA library. Mutating the intron sites of the *CALR* gene by base editing significantly inhibited JEV replication. Furthermore, H3K4me3 and H3K27ac histone signals were detected in this region, indicating that this intron mutation location might be a transcription factor binding site.

Error-prone PCR and DNA editors are no longer suitable for mutation generation and protein evolution studies due to their inherent inefficiency and complexity [9,25,26]. The development of CRISPR technology has facilitated progress in the field of protein evolution [27,28,29]. The CRISPR-Cas9 system has been applied to induce insertions and deletions (indels) in the target sites for generating targeted mutations [30]. However, mutations produced by CRISPR-Cas9 are typically nonfunctional or out-of-frame mutations are unsuitable for directed evolution. In addition, the double-strand breaks (DSBs) triggered by Cas9 have aroused public concern about the safety of this system [31,32]. The CRISPR-mediated base editor can produce specific mutations without inducing DSBs, showing better security and higher efficiency, which is more conducive to protein evolution in basic functional genomics research [10,33]. Previous studies on base editing screening of critical genes primarily focused on gene exon functions overlooking the intron regions [13,14]. Hence, this study optimized the sgRNA library covering all the exons and introns of the *CALR* gene and realized the saturation coverage of the whole gene sequences.

Notably, the intron mutation identified in this study was related to virus replication, consistent with previous reports that some intron regions might be associated with virus infection [34]. The top two most enriched sgRNAs screened in this study were in the matching segment of the genome, further confirming the trustworthiness of this mutation site. Compared with the wild-type cells, the Mut-1 cell line with this intron mutation significantly inhibited the replication of JEV, but this cell line still expressed normal levels of the CALR protein, which was partially inconsistent with one previous report that knocking out *CALR* inhibited JEV replication [5]. Such inconsistency might be due to the influence of transcription factor binding or remote regulation. However, the mechanism by which intron mutation inhibits JEV replication needs further exploration.

We successfully constructed an sgRNA library covering all the exons and introns of the *CALR* gene. Previous genome-wide knockout screening studies of human infectious diseases identified critical target genes. This method is limited to gene screening and cannot be used for identifying specific mutations, whereas the saturation mutation screening method developed in this study can be applied for exploring exact domains or regions of the target genes [5,35,36]. Therefore, our technique is conducive to revealing the unknown function and mechanism of the target gene. Since our method requires no gene knockout, it can avoid possible embryonic death induced by gene knockout and thus provide new resources for disease models.

Nevertheless, there are still some limitations in our research. First, the types of mutations generated by the system are limited to the editing window and types (C to T) [10]. Secondly, we failed to completely cover all the sequences of the *CALR* gene due to the PAM limitation of sgRNA. Future studies can break through these limitations by applying prime editing, SpRY-based base editing or dual-functional base editing systems to increase the types of mutations [37,38,39,40]. Overall, this study reveals the potential mechanism by which intron mutations inhibit JEV replication. In addition to screening critical mutations, the developed method can identify key disease-resistant targets.

## Figures and Tables

**Figure 1 ijms-24-13331-f001:**
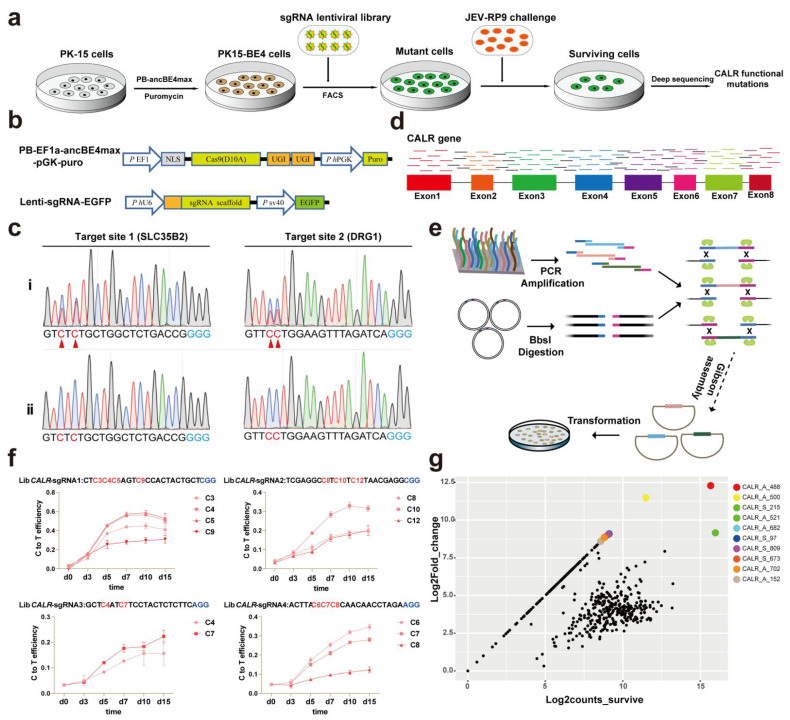
Schematic of identifying critical mutations of the *CALR* gene. (**a**) Workflow and screening procedures by the CRISPR-mediated base editing screening. (**b**) Structure elements of the PiggyBac vector (PB-EF1ɑ-ancBE4max-pGK-puro) and the sgRNA expression vector (Lenti-sgRNA-EGFP). (**c**) Sequencing analysis of two target sites (*SLC35B2* and *DRG1*). Red triangles indicate edited nucleotides. (**i**) Denotes the treatment group; (**ii**) Denotes the control group. (**d**) Schematic of the *CALR* gene showing the 8 exons, 7 introns and the 457 specific sgRNAs. The coloured squares and black lines linking squares indicate exons and introns, respectively. (**e**) The pipeline of sgRNA library construction. (**f**) Editing efficiency of 4 sgRNAs randomly selected from the designed sgRNA library in PK15-BE4 cell line. The editing efficiency was calculated by the editR program. The data were expressed as the means ± SD of three independent experiments. (**g**) Scatter dot plots showing sgRNA frequencies and enrichment extent. The viable cells were compared between the JEV infection group and the noninoculated control group.

**Figure 2 ijms-24-13331-f002:**
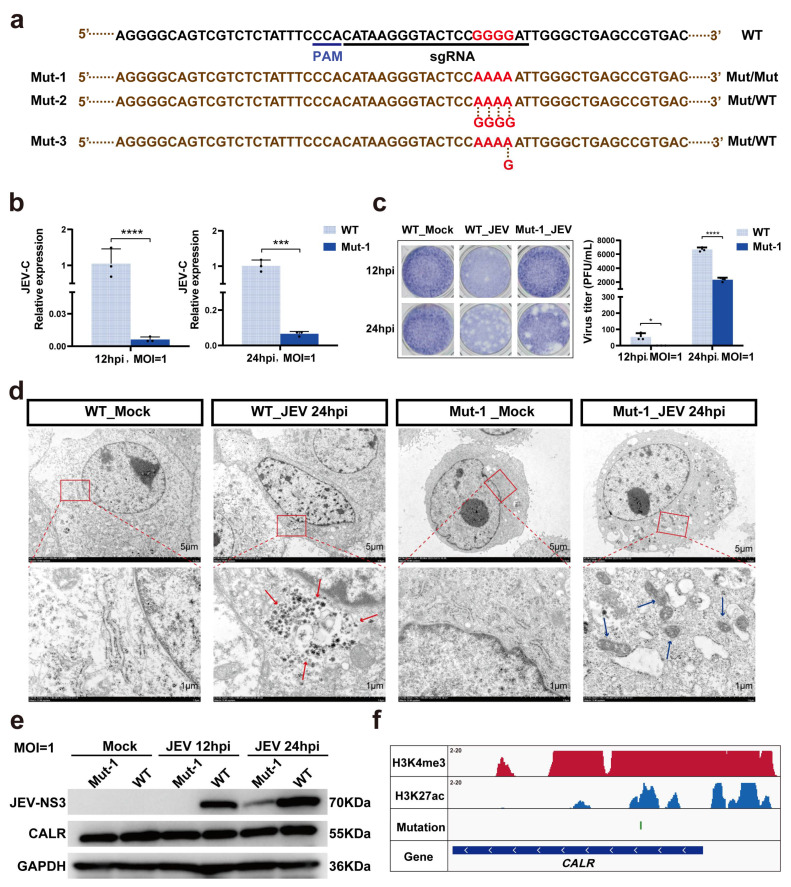
Significant inhibition of JEV replication by intron mutation. (**a**) Genotypes of cell lines edited by *CALR*_A_521 sgRNA. Mut-1 refers to homozygotes; Mut-2 and Mut-3 refer to heterozygotes. (**b**) Expression of JEV-C gene in JEV-infected Mut-1 cells detected by qRT-PCR. (**c**) Viral concentration in Mut-1 cells by a virus plaque assay. (**d**) Effects of intron mutation (Mut-1) on virus particle assembly by negative-staining electron microscopy. Red arrows point to scattered virus particles in the endoplasmic reticulum. Blue arrows point to the mitochondria of Mut-1 cells infected with JEV. Scale bar, 5 µm or 1 µm. (**e**) Western blot analysis of *CALR* and JEV-NS3 protein in Mut-1 and WT cells. (**f**) CUT&Tag analysis of the region where the intron mutation occurred in PK-15 cells. PAM, protospacer adjacent motif; JEV, Japanese encephalitis virus; MOI, multiplicities of infection; hpi, hours post-infection; WT, wild-type; kDa, kilodalton. Non-infected cells were used as negative control (Mock). Data are expressed as means ± SD (n = 3). * *p* < 0.05; *** *p* < 0.001; **** *p* < 0.0001. Two-tailed Student’s *t*-test was performed to determine statistically significant differences among groups.

## Data Availability

The data related to this paper may be requested from the authors.

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
