# Peer review of "CRISPR-Cas9-Mediated Cytosine Base Editing Screen for the Functional Assessment of CALR Intron Variants in Japanese Encephalitis Virus Replication"

_ijms, 2023, doi:10.3390/ijms241713331_

Round 1

Reviewer 1 Report

I have confirmed many related studies since the CRISPR-Cas9 technology came out, and it seems that appropriate contents are included in the paper. This is expected to be helpful for research related to JEV derived from CALR intron variants. However, it is not considered appropriate for the journal level of IJMS (IF=5.6).

1. Some typo errors are identified. Typically, "in vitro" should be written in italics. Please check the manuscript once again as a whole.

2. The order of the Supplementary Tables does not match at all. Please rearrange them in consideration of the order in the text.

3. There is no problem with the logical flow in the overall content, and the amount is appropriate.

Author Response

Comment 1:

Some typo errors are identified. Typically, "in vitro" should be written in italics. Please check the manuscript once again as a whole.

Reply 1:

Thanks for your remind. This has been acknowledged and the term “in vitro” has been italicized in the manuscript document.

Comment 2:

The order of the Supplementary Tables does not match at all. Please rearrange them in consideration of the order in the text.

Reply 2:

Thanks for your remind. The supplementary tables have been reordered and clearly annotated.

Comment 3:

There is no problem with the logical flow in the overall content, and the amount is appropriate.

Reply 3:

Thanks for your suggestion. The same order of logical flow has been maintained in the revised manuscript.

Reviewer 2 Report

In this brief report,  Xiong and colleagues use CRSPR-Cas9 technology to perform functional analysis of the Calreticuln (CALR) gene mutants on replication of Japanese encephalitis Virus (JEV replication). A four G consecutive repeat sequence in intron 2 of the CALR gene was found to inhibit replication of JEV in 

Comments:  

1.      The focus on the abstract and intro is more on CRSPR-Cas9 system rather than the specific relevance of identifying determinants of JEV replication. More emphasis should be placed on JEV

2.     Figure fonts should be larger—difficult to read in printed copy 

3.     Figure 1 F, statistical analysis should be shown on graph. Statistical methods are not adequately described and p values should be shown. 

Author Response

Comment1:

The focus on the abstract and intro is more on CRSPR-Cas9 system rather than the specific relevance of identifying determinants of JEV replication. More emphasis should be placed on JEV.

Reply 1:

Thanks for your suggestion. More information related to JEV replication and the significance of identifying JEV determinants are included in Lines 21-26 and Lines 47-52.

Comment 2:

Figure fonts should be larger—difficult to read in printed copy.

Reply 2:

We have bolden and increased the font size in the images. In addition, we have uploaded high resolution supplementary files together with the manuscript.

Comment3:

Figure 1 F, statistical analysis should be shown on graph. Statistical methods are not adequately described, and p values should be shown.

Reply 3:

Thank you for your suggestion. We are very sorry that we did not describe it clearly in the manuscript. The purpose of this experiment is to evaluate the editing efficiency of library sgRNA in the PK15-BE4 cell line at different time points, and determine the optimal screening time for the following experiments. The revised description of this result is included in Lines 250-252.
